# Multi-Omic Investigations of a 17–19 Translocation Links *MINK1* Disruption to Autism, Epilepsy and Osteoporosis

**DOI:** 10.3390/ijms23169392

**Published:** 2022-08-20

**Authors:** Jesper Eisfeldt, Jakob Schuy, Eva-Lena Stattin, Malin Kvarnung, Anna Falk, Lars Feuk, Anna Lindstrand

**Affiliations:** 1Department of Molecular Medicine and Surgery, Karolinska Institutet, 171 76 Stockholm, Sweden; 2Department of Clinical Genetics, Karolinska University Hospital, 171 76 Stockholm, Sweden; 3Science for Life Laboratory, Karolinska Institutet Science Park, 171 65 Solna, Sweden; 4Department of Immunology, Genetics and Pathology, Uppsala University, 751 08 Uppsala, Sweden; 5Department of Neuroscience, Biomedicum, Karolinska Institutet, 171 77 Stockholm, Sweden; 6Lund Stem Cell Center, Department of Experimental Medical Science, Lund University, 221 84 Lund, Sweden; 7Science for Life Laboratory, Uppsala University, 752 37 Uppsala, Sweden

**Keywords:** reciprocal translocation, *MINK1*, autism, epilepsy, osteoporosis, long-read genome sequencing, transcriptome sequencing, patient-specific neural stem cells

## Abstract

Balanced structural variants, such as reciprocal translocations, are sometimes hard to detect with sequencing, especially when the breakpoints are located in repetitive or insufficiently mapped regions of the genome. In such cases, long-range information is required to resolve the rearrangement, identify disrupted genes and, in symptomatic carriers, pinpoint the disease-causing mechanisms. Here, we report an individual with autism, epilepsy and osteoporosis and a de novo balanced reciprocal translocation: t(17;19) (p13;p11). The genomic DNA was analyzed by short-, linked- and long-read genome sequencing, as well as optical mapping. Transcriptional consequences were assessed by transcriptome sequencing of patient-specific neuroepithelial stem cells derived from induced pluripotent stem cells (iPSC). The translocation breakpoints were only detected by long-read sequencing, the first on 17p13, located between exon 1 and exon 2 of *MINK1* (Misshapen-like kinase 1), and the second in the chromosome 19 centromere. Functional validation in induced neural cells showed that *MINK1* expression was reduced by >50% in the patient’s cells compared to healthy control cells. Furthermore, pathway analysis revealed an enrichment of changed neural pathways in the patient’s cells. Altogether, our multi-omics experiments highlight *MINK1* as a candidate monogenic disease gene and show the advantages of long-read genome sequencing in capturing centromeric translocations.

## 1. Introduction

Genomic reciprocal translocations form a subgroup of balanced structural variants (SVs) where genetic segments are exchanged between non-homologous chromosomes. Most commonly, the translocations are balanced events in terms of copy number state and are, therefore, normally not pathogenic. However, when the breakpoints (BPs) occur within genes or gene regulatory segments, the transcripts may be affected, leading to phenotypic consequences [1,2]. In genetic diagnostics, reciprocal translocations are still mostly detected by karyotyping of metaphase chromosomes. Furthermore, we and others have successfully applied short-read genome sequencing (srGS) to detect some translocations [2,3]. However, the detection of balanced SVs such as translocations is relatively poor compared to other variant classes [4]. The difficulty of pinpointing balanced SVs is largely due to the short fragment length of the srGS libraries, complicating the alignment within repetitive genomic regions [5].

A wide range of technologies has been developed to overcome these shortcomings, including linked-read genome sequencing (liGS) [6], long-read genome sequencing (lrGS) [7], as well as optical mapping [8]; these methods utilize different technological approaches to characterize the sequence or structure of high molecular weight (HMW) DNA. Briefly, liGS constitutes library preparation methods that allow for the separation of HMW DNA into oil droplets acting as individual reaction chambers and subsequent DNA fragmentation. These fragments are labelled using a molecular barcode specific to each HMW DNA molecule. Lastly, the barcoded fragments are sequenced using an srGS platform, allowing for in silico reconstruction of the HMW DNA molecules [9]. LrGS consists of multiple technologies capable of sequencing HMW DNA directly [10], generating reads that typically are longer than 10 Kbp; to date, lrGS is relatively expensive and inaccurate when compared to srGS. Optical mapping is used to generate high-resolution maps across the entire genome. The optical maps are generated by labeling specific DNA motives with fluorescent probes. Once the HMW DNA molecules are labeled, the molecules are stretched, and the fluorescent probes are imaged. Lastly, the distance between the fluorescent probes is scanned and compared to a reference genome, allowing for the detection of SVs [11]. Compared to srGS and lrGS, optical mapping is generally used to analyze longer fragments of DNA at a lower resolution and lower cost. Although there is a plethora of comparisons and case reports showcasing the value of these technologies [7,12], it is yet unclear which method is the most suitable for detecting balanced SVs, such as centromeric translocations.

In this study, we describe the clinical details of a patient with a de novo balanced reciprocal translocation between chromosomes 17 and 19 and compare different genomic technologies to investigate the structural and transcriptomic effects of the detected SV. The translocation breakpoint junctions (BPJs) were identified by only one out of four technologies, lrGS, on the Oxford Nanopore PromethIon platform, which further showed that *MINK1* (Misshapen-like kinase 1) was disrupted. By transcriptome analysis of patient-specific neuroepithelial stem cells (NESCs) derived from induced pluripotent stem cells (iPSCs), we then demonstrated a reduction of *MINK1* expression levels. Altogether, our multi-omics experiments highlight *MINK1* as a candidate monogenic disease gene and show the advantages of lrGS in capturing centromeric translocations.

## 2. Results

### 2.1. Clinical Findings

The male patient was born with a congenital cataract of the left eye and had a history of epileptic seizures as a child. Chromosome analysis revealed a de novo balanced translocation between chromosome 17 and 19, 46,XY,t(17;19) (p13;p11) (Figure 1A).

At age 32, he had a diagnosis of high functioning autism and suffered from recurrent depression. Skeletal radiology after a traumatic event of unknown cause revealed multiple fractures (bilateral humeral neck fractures with dislocation of the shoulders, vertebral compression fractures and a sternum fracture), and subsequently, a diagnosis of osteoporosis was established. Finally, the patient had a pathological electroencephalogram (EEG), which showed epileptic activity. There was no history of seizures in adulthood, but it is likely that the above-mentioned traumatic event was secondary to a generalized tonic-clonic seizure.

### 2.2. Genomic Analyses

BPJ mapping was attempted utilizing srGS (Illumina PCR-Free GS; Figure 1B; Illumina, San Diego, CA, USA), liGS (10X chromium liGS; Figure 1B) and optical mapping (Bionano genomics Saphyr, Appendix A, Bionano Genomics, San Diego, CA, USA). However, the translocation was not detected in any of those analyses, making further characterization impossible by any of these techniques.

We then performed lrGS sequencing on the Oxford Nanopore platform and were able to resolve the rearrangement, with the der(17) BP disrupting *MINK1* between exon 1 and 2 and the der(19) BP located inside the centromere (Figure 1B,C). Investigation of the chromosome breakpoint region in short and liGS data revealed informative reads (Figure 1B) that were correctly mapped within *MINK1* but failed to align uniquely to the chromosome 19 centromere. The srGS data indicated a t(1;17) translocation, while the molecules of the liGS data aligned to the decoy contig. There was no sign of translocation in the Bionano optical mapping data.

Next, we aligned the BPJ sequences against GRCh37 and GRCh38. Interestingly, in GRCh37, the der(17) harbored a random insertion of 26 nucleotides (nt), but in GRCh38, a blunt end was observed, and the 26 nt were correctly mapped to the chr19 centromere. In the der(19) BPJ, two random insertions of 17 and 5 nt separated by 8 nt were present in both reference genomes.

### 2.3. RNA-Seq of Neural Stem Cells

Next, we were interested in analyzing what effect the translocation had on the expression of *MINK1* in disease-relevant patient-specific neural cells, and we performed transcriptome sequencing of the patient’s NESCs and three unrelated controls. Through these analyses, we discovered a total of 539 differentially expressed genes (Appendix A), including *MINK1* (*p* = 1.1 × 10^−11^, log2fold change = −0.92) (Figure 2A). The differential expression of *MINK1* was validated using qPCR (Appendix A), revealing significant downregulation compared to healthy control cells (Figure 2A). Analyzing *MINK1* in GTEx (https://www.gtexportal.org/home/gene/MINK1, accessed on 21 January 2022) [13]*,* we found that *MINK1* is abundantly expressed in tissues relevant to the patient’s phenotype, such as the cerebellum (median TPM 142.6) and the cortex (median TPM 89.25) (Figure 2B). Except for *MINK1*, there were no differentially expressed genes within 1 Mbp of the translocation breakpoints.

Allele-specific expression analyses of *MINK1* were not possible due to the lack of informative heterozygous SNVs, and no aberrant *MINK1* transcripts were found, indicating that the affected allele is degraded through nonsense-mediated decay.

The differential expression analysis was followed up with STRING protein–protein interaction analysis [14] (Appendix A), revealing 975 protein interactions between the 539 differentially expressed genes, representing a significant enrichment of such interactions (*n* = 709, *p* = <1.0 × 10*^−^*^16^), indicating that these proteins are biologically connected. In particular, we found that 70% of the differentially expressed genes (376 of 539) were part of a protein–protein interaction network together with MINK1.

Next, we analyzed the network using Panther GO biological process enrichment analysis (Appendix A) [15]. The GO enrichment analysis resulted in the discovery of 365 enriched biological processes, including phenotypically relevant terms such as pathway-restricted SMAD protein phosphorylation (GO:0060389), ossification (GO:0001503) and nervous system development (GO:0007399), as well as seemingly unrelated terms, including ovulation cycle (GO:0042698) and female pregnancy (GO:0007565); indicating that the *MINK1* differentially expressed gene network is involved in a wide range of biological functions.

## 3. Discussion

In this study, we investigated, at the molecular level, a reciprocal translocation between chromosomes 17 and 19. We compared four genomic technologies, Illumina srGS, liGS, Bionano optical mapping and Nanopore lrGS. Notably, Nanopore lrGS was the only technology able to detect the translocation, which is remarkable given the low coverage of the Nanopore sequencing data (3X). To the best of our knowledge, this is the first time lrGS has been utilized to capture a translocation with a centromeric breakpoint of a patient sample. Our results indicate that shallow lrGS is sufficient for detecting SV, even in repetitive regions and could serve as a cost-efficient complement to srGS in clinical settings. Shallow lrGS could fulfill a variety of purposes in the diagnostic pipeline, including the detection of repeat expansions and the phasing of SNVs.

Analyzing the breakpoints of the translocation, we noticed that there were clear translocation signatures in the srGS and liGS datasets (Figure 1C), indicating that the centromeric translocation could be detected in these datasets using appropriate software. In particular, we envision a software algorithm that searches for clusters of read pairs, such that one read is located in non-centromeric regions, and their mates align within centromeric regions. To our knowledge, there is yet no srGS caller available for such analyses; however, there are similar tools for the discovery of mobile element insertions [16].

Comparing the performance of the srGS alignment with the liGS alignment, we found that the srGS data reports a centromeric chromosome 1 translocation; since we were actively searching for a t(17;19) translocation, this finding was discarded as a false positive; on the other hand, the liGS dataset reported an insertion of material from the decoy contig, which is complicated to interpret and unlikely to result in the discovery of a true SV. Therefore, in lieu of previously published studies [12], we found that the decoy contig is detrimental to the detection rate of balanced SV.

The optical mapping data did not reveal any signs of the translocation, which is remarkable, given the great depth (>70X) and length of the optical mapping datasets (298 Kbp average molecule length). Notably, there were no chimeric contigs aligned within *MINK1*, indicating that the analysis pipeline failed to assemble the breakpoint junction. Therefore, it could be valuable to develop a mapping assembly approach for the optical mapping data as a complement to the de novo assembly approach used by the Bionano Solve pipeline.

Analyzing the breakpoint sequences of the translocation breakpoint, we discovered the presence of large non-templated insertions (Figure 1D); however, when aligning the data to GRCh38, one of the non-templated insertions was shown to be a reference sequence missing from GRCh37. As such, there are practical benefits in transitioning from GRCh37 toward GRCh38. With the recent release of the telomere-to-telomere assembly T2T-CHM13 [17], the centromeric regions are fully sequenced using a combination of sequencing methods, mostly based on lrGS. We acknowledge the advantage of such a reference when facing genomic rearrangements that include repetitive segments such as centromeres and telomeres. Alternatively, we argue that the greatest benefit would come by a transition toward a graph genome; preferably, such graph a genome should be based on lrGS data from multiple individuals in the local population.

Altogether, the rearrangement reported here may have occurred by a replicative mechanism such as fork-stalling and template-switching (FoSTeS)/microhomology-mediated break-induced replication (MMBIR) where microhomology and short stretches of nucleotide insertions are frequently observed in the BPJs [18]. However, due to the shortcomings of the reference genome in the centromere, we cannot truly characterize the features in the BPJs illustrated by the der(17) insertion, actually representing sequences missing from the GRCh37 reference as discussed above. Hence, we refrain from suggesting a mechanism of formation in this case.

Through the analysis of the Nanopore lrGS data, we discovered that the translocation breakpoint disrupted gene *MINK1*, and through follow-up studies in iPSC, we revealed that *MINK1* is downregulated in neural cells compared to healthy control cells. Through database searches, we found that *MINK1* has a pLI score of 1 [19], indicating that the loss of *MINK1* is not tolerated. Our hypothesis is, therefore, that *MINK1* haploinsufficiency is causative of the clinical features seen in the patient. It remains to be elucidated if all or only some of these features have a common etiology. However, we suggest *MINK1* as a novel candidate gene for autism, congenital cataract, epilepsy and osteoporosis. Although we report the first patient with disruption of *MINK1*, the gene has already been implicated in a variety of disorders based on cohort studies in humans, in silico models and work on model organisms, including congenital heart disease in humans [20] and skeletal and neuronal phenotypes in mice [21,22], as well as cancer [23,24], platelet formation [25], Alzheimer disease [26,27] and arthritis [28]. As such, it appears that *MINK1* is part of a diverse set of biological processes, which is expected as *MINK1* is highly expressed in a variety of tissues (Figure 2B) and interacts with a diversity of proteins (Appendix A). Searching the GWAS central for traits and diseases associated with *MINK1*, we found that *MINK1* has been associated with partial epilepsy [29], as well as other NDD traits, including Bipolar disorder, Brain glutamate concentrations and Schizophrenia [30]. Utilizing GTEx and STRING to study these functions in more detail, we found that *MINK1* is highly expressed in the brain, supporting a role in NDD disease (Figure 2B).

## 4. Materials and Methods

### 4.1. Genome Analysis

In brief, genomic DNA derived from whole blood was sequenced to 30X depth at the National Genomics Infrastructure (NGI), Stockholm, Sweden, using the PCR-free paired-end (PE) protocol for srGS and the 10X Genomics Chromium GS protocol for liGS. The Data were processed and analyzed as described previously [12].

Oxford Nanopore LrGS was performed at the National Genomics Infrastructure (NGI) Uppsala. The sequencing was performed using the Promethion platform, and bases were called using the Guppy base caller (https://nanoporetech.com, accessed on 18 January 2022), producing 838,235 reads, with an average length of 10 Kbp resulting in roughly 3X coverage. The resulting Nanopore lrGS data were aligned to GrCh37 using Minimap2 [31], and variants were called using Sniffles [7], setting the minimum read support parameter to three reads.

Optical mapping was performed on a genomic DNA sample from the proband by running dual enzymes (BspQI, BssSI) on the Bionano Genomics (San Diego, CA, USA) Saphyr platform (https://bionanogenomics.com/support-page/saphyr-system, accessed on 8 December 2021). Analysis was performed as described previously [12]. Briefly, the optical maps were analyzed using Bionano-solve (https://bionanogenomics.com/support-page/bionano-solve, accessed on 24 January 2022), aligned to GrCh37 reference genome using Bionano RefAligner (version 5649) and output files were converted into VCF files using a custom script (https://github.com/J35P312/smap2vcf, accessed on 26 January 2022). Variants of interest were visualized in Bionano access.

Identified BPJs were validated by breakpoint PCR, as described previously [12]. Both PCR sequences, as well as the informative soft clipped reads from the lrGS data, were aligned to both GrCh37 and GrCh38 using BLAT (UCSC Genome Browser) [32].

### 4.2. Neural Stem Cell Cultivation and Transcriptome Sequencing

NESCs were grown at the iPS Core facility at Karolinska Institutet, according to standard protocols (https://ki.se/en/research/ips-core-facility, accessed on 7 December 2021). In brief, NESCs were established from iPSC of reprogrammed fibroblast cells, sampled from the patient, as well as three healthy unrelated control individuals. Transcriptome sequencing was performed on each of these four individuals at NGI Stockholm, using a Ribozero-prep, sequencing 44 million 150 bp paired reads per replicate (three replicates per individual) on the Nova-seq platform. The resulting data were aligned to GrCh37 using STAR and Salmon [33]. Differential expression analysis was performed using Deseq2 [34] with the default settings. The tests for statistical significance were included in the DeSeq2 package and employed the Wald test with subsequent *p*-value correction by the Benjamini–Hochberg technique.

## 5. Conclusions

In conclusion, we demonstrated the usefulness of lrGS in diagnostic settings by detecting a centromeric translocation. We predict that such technology will likely be useful in the clinic in the future for selected samples. Here, the ability to pinpoint the genomic breakpoints unraveled a gene disruption. Even though it is a single case, several layers of evidence from both induced patient neural cells as well as data mining of public databases indicate that the *MINK1* disruptions are the likely explanation for autism, epilepsy and osteoporosis in the patient reported here. In patient-derived neural cells, we observed reduced *MINK1* expression. In RNA-Seq data from the same cells, we detected changes in cellular pathways related to the patient’s phenotypes. Finally, we confirmed that *MINK1* is important in neural development by data mining of public datasets.

## Figures and Tables

**Figure 1 ijms-23-09392-f001:**
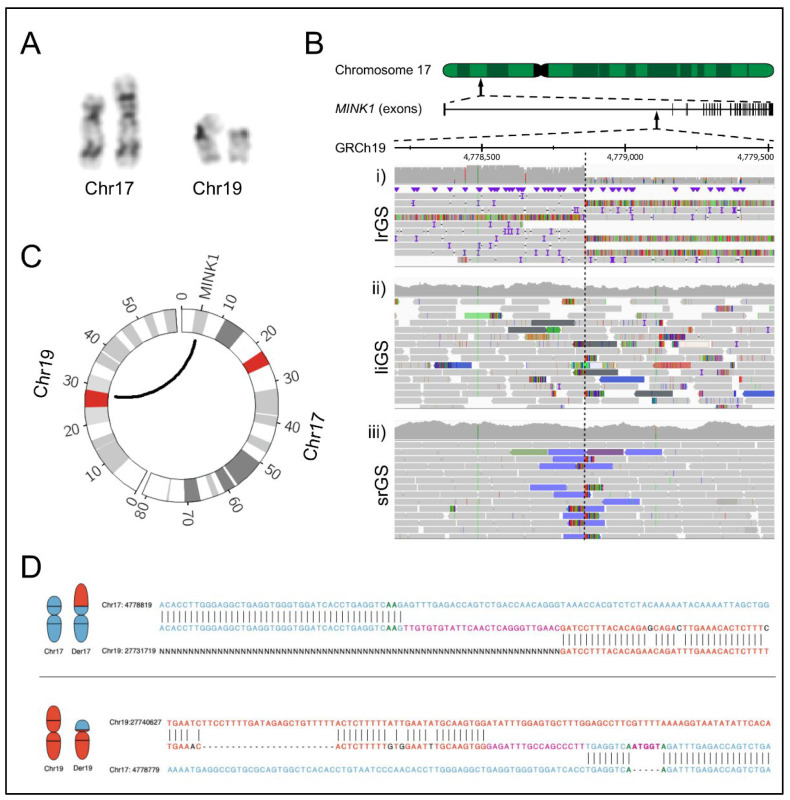
Genomic analyses of the t(17;19) translocation. (**A**) Excerpt of the patient karyogram, indicating a t(17:19) translocation. (**B**) View in the Integrative Genomics Viewer (IGV) of the translocation breakpoint in (i) Oxford Nanopore GS data (lrGS), (ii) 10X Chromium GS data (liGS), (iii) Illumina PCR-free GS data (srGS). (**C**) A circos plot illustrating the exact breakpoints of the translocations. (**D**) The upper pane illustrates the breakpoint junction sequence of Der (17), while the lower pane illustrates the sequence of Der (19). The reference sequences are shown as the lower and upper sequences in each pane, while the junction sequence is shown between. Horizontal lines indicate a match, purple sequence illustrates insertion. The sequence of chr19 is in red, and the sequence of chromosome 17 in blue.

**Figure 2 ijms-23-09392-f002:**
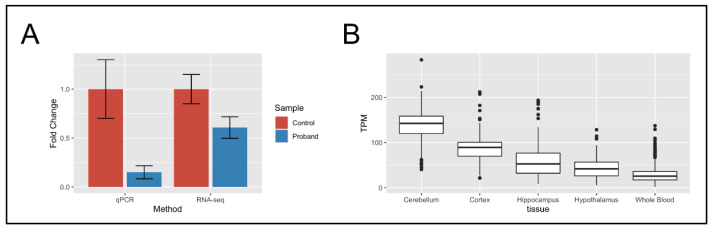
Downregulation and expression of MINK1. (**A**) Expression data from qPCR and RNA-Seq showed the downregulation of MINK1 in the proband NESCs compared to healthy control individuals. (**B**) Expression data of MINK1 in healthy individuals, as seen in GTEx, showed that the gene is higher expressed in neural tissue compared to blood, which highlights the importance of appropriate RNA sampling for relating transcriptomic analysis to clinical data. TPM, transcripts per million.

## Data Availability

All data supporting the conclusions in the article are presented in the main text or supplementary data files.

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
