# Peer review of "Multi-Omic Investigations of a 17–19 Translocation Links MINK1 Disruption to Autism, Epilepsy and Osteoporosis"

_ijms, 2022, doi:10.3390/ijms23169392_

Round 1
Reviewer 1 Report
In this manuscript, Eisfeldt et al performed a beautiful study to identify the association of Mink1 to autism, epilepsy and osteoporosis. I speculate that the study would generate a lot of interest and may generate significant therapeutic values in future. Overall, I think this is a decent study that is thoroughly designed, thoughtfully executed and nicely presented. By answering the following issues, I think this would be a great addition to MDPI-IJMS.
(1) The authors should do gene expression changes of few other genes in that locus to make sure other genes in that region are not significantly changed. They are claiming that it is a “monogenic” disorder, so this point is particularly important.
(2) The authors need to show a western blot also to show that it is not only the RNA of MINK1, the protein levels are also reduced in this patient.
(3) Can they test the expression of MINK1 in other tissues other than neuronal samples? Technicality could be an issue for the assay, but this could at least be tested easily by getting blood sample from the patient.
(4) Finally, for a strong conclusion, getting evidence from only one patient may not be enough to make a big claim. I can understand it may be tough sometime to get samples from more patients, but can the authors at least refer some publications or GWAS studies where MINK1 is associated with diseases similar to the one investigated in this study?
Author Response
Comments and Suggestions for Authors
In this manuscript, Eisfeldt et al performed a beautiful study to identify the association of Mink1 to autism, epilepsy and osteoporosis. I speculate that the study would generate a lot of interest and may generate significant therapeutic values in future. Overall, I think this is a decent study that is thoroughly designed, thoughtfully executed and nicely presented. By answering the following issues, I think this would be a great addition to MDPI-IJMS.
Thank you for this positive response to our study!
(1) The authors should do gene expression changes of few other genes in that locus to make sure other genes in that region are not significantly changed. They are claiming that it is a “monogenic” disorder, so this point is particularly important.
Thank you for this relevant comment. On chr 17, 52 genes were located within 1 million base pairs (mbp) of the breakpoint junction and none of them were differentially expressed. No genes were located within 1mbp of the junction on chromosome 19. This has been added to the article results (lines 121-122).
(2) The authors need to show a western blot also to show that it is not only the RNA of MINK1, the protein levels are also reduced in this patient.
We appreciate the reviewer’s suggestion. In this study we chose to focus on the multiple OMICs analysis performed in this unique case.
Although further studies on the translational level are worth pursuing, we believe that this is beyond the scope of this case report. Further studies of the gene in the generated patient neural cells (and even in mixed population neurons) would be very interesting but in order to be truly informative, additional patient samples and/or generated loss of function lines (e.g. CRISPR/CAS9) would be needed to match with the patient cells.
(3) Can they test the expression of MINK1 in other tissues other than neuronal samples? Technicality could be an issue for the assay, but this could at least be tested easily by getting blood sample from the patient.
Although the reviewer makes a valid point, further experiments investigating the expression of MINK1 in additional tissues would be interesting, for the individual studied here we don’t have access to additional primary samples. Therefore, follow up experiments would need to be done on differentiated NESCs (e.g. neurons) as well as the iPSCs which the NESCs originate from. As in (2) above, it would be more interesting to differentiate the cells further into neurons or glia cells or include secondary model like organoids and study the expression and protein levels there. However, we believe that this is beyond the scope of this report.
(4) Finally, for a strong conclusion, getting evidence from only one patient may not be enough to make a big claim. I can understand it may be tough sometime to get samples from more patients, but can the authors at least refer some publications or GWAS studies where MINK1 is associated with diseases similar to the one investigated in this study?
Thank you! We have mined the GWAS central and found association to both epilepsy and neurological disorders. This has been added to the main text (lines 225-228).
Reviewer 2 Report
The article is excellent, showing a successful approach of the authors to identify MINK1 as a candidate gene for the disease of a patient with autism, epilepsy and osteoporosis bearing de novo balanced t(17;19) by virtue of long-read genome sequencing technique and a technique for NESC differentiation of patients-derived iPSCs. Although this is a case report, their success will encourage researchers to take this strategy to identify responsible genes for other neurological diseases and acculturated information will contribute to identifying a new mechanism for the development of various neurological disorders. The current work is worth reporting since it triggers such movements.
The reviewer recommends this article as a strong candidate for publication in IJMS.
Author Response
Thank you for this positive review of our article.
Round 2
Reviewer 1 Report
I agree with the revision from the authors and believe that the revised version is improved. I would still urge the authors to do the western blots as soon as possible and may be add that data to their next publication related to this. For any genetic study, this is a critical step to validate the predicted mechanism. Otherwise, congratulations to the team for a nice piece of work!